A method to adjust a prior distribution in Bayesian second-level fMRI analysis

Han Hyemin hyemin.han@ua.edu
Educational Psychology Program, University of Alabama - Tuscaloosa , Tuscaloosa , AL , United States of America
Gollo Leonardo
Electronic publication date: 2021 Feb 3
Publication date: 2021
Volume: 9
Electronic Location ID: e10861
Received 2020 Oct 2; Accepted 2021 Jan 8
Copyright: ©2021 Han
Copyright year: 2021
Copyright holder: Han
License: This is an open access article distributed under the terms of the Creative Commons Attribution License, which permits unrestricted use, distribution, reproduction and adaptation in any medium and for any purpose provided that it is properly attributed. For attribution, the original author(s), title, publication source (PeerJ) and either DOI or URL of the article must be cited.
License URL: https://creativecommons.org/licenses/by/4.0/

Keywords: Bayesian analysis, Cauchy distribution, fMRI, Prior distribution, Meta-analysis, Bayes factor

Funding: The author received no funding for this work.

==============================
Previous research has shown the potential value of Bayesian methods in fMRI (functional magnetic resonance imaging) analysis. For instance, the results from Bayes factor-applied second-level fMRI analysis showed a higher hit rate compared with frequentist second-level fMRI analysis, suggesting greater sensitivity. Although the method reported more positives as a result of the higher sensitivity, it was able to maintain a reasonable level of selectivity in term of the false positive rate. Moreover, employment of the multiple comparison correction method to update the default prior distribution significantly improved the performance of Bayesian second-level fMRI analysis. However, previous studies have utilized the default prior distribution and did not consider the nature of each individual study. Thus, in the present study, a method to adjust the Cauchy prior distribution based on a priori information, which can be acquired from the results of relevant previous studies, was proposed and tested. A Cauchy prior distribution was adjusted based on the contrast, noise strength, and proportion of true positives that were estimated from a meta-analysis of relevant previous studies. In the present study, both the simulated images and real contrast images from two previous studies were used to evaluate the performance of the proposed method. The results showed that the employment of the prior adjustment method resulted in improved performance of Bayesian second-level fMRI analysis.

Introduction

fMRI (functional magnetic resonance imaging) has been widely used by neuroscientists and psychologists who are interested in examining the neural-level mechanisms of psychological and behavioral processes of interest that could not be well investigated by traditional research methods in the field, such as surveys, observation, interviews, etc (Logothetis, 2008; Han, 2016). fMRI shows localized increase of blood volume that can be used as a proxy for neural-level functioning (Bandettini, 2012). With this information, researchers are able to associate psychological and behavioral processes with localized brain functioning by examining which brain regions are showing increased blood volume during specific behaviors or mental processes (Logothetis et al., 2001). One of the major methodological benefits of fMRI is that it allows us to observe the neural correlates of the processes of interest in a non-invasive way, unlike other neuroscientific research methods (Logothetis, 2008). It also provides good spatial resolution, which is required for the localization of the processes of interest at the neural level (Mulert et al., 2004). As a result, fMRI methods have been employed in many subfields of psychology and neuroscience ranging from cognitive psychology to social neuroscience (Han, 2016).

Notwithstanding the aforementioned benefits of fMRI methods, several fMRI researchers have expressed their concerns regarding the performance of the methods, such as thresholding with multiple comparison correction, implemented in widely-used analysis tools, such as SPM, FSL, AFNI, etc (Han & Glenn, 2018; Han, Glenn & Dawson, 2019). For example, Woo, Krishnan & Wager (2014) and Eklund, Nichols & Knutsson (2016) have shown that use of default analysis options provided in the tools can lead to inflated false positives, which could be a significant issue in fMRI analysis involving multiple simultaneous tests. As a possible alternative approach in fMRI analysis, Bayesian analysis has been suggested in recent studies (Han & Park, 2018; Han, 2020b). Studies have shown that the employment of Bayesian analysis in fMRI analysis, particularly second-level analysis and meta-analysis (Han & Park, 2019), resulted in significantly improved performance. They reported that compared with classical frequentist analysis with multiple comparison correction, Bayesian analysis showed significantly improved sensitivity while maintaining reasonable selectivity (e.g., false-positive rate < .05) when voxelwise inference was performed.

In addition to this practical benefit, research has suggested the epistemological benefit of use of Bayesian analysis as well. A widely used indicator for frequentist inference, a p-value, in fact cannot be used to examine whether an alternative hypothesis of interest is supported by data (Wagenmakers et al., 2018b). p-values are mainly about the extremity of the distribution of observed data given a hypothesis, not about to what extent the data supports the hypothesis (Cohen, 1994). Hence, p-values can only indicate whether a null hypothesis can be rejected but not whether an alternative hypothesis can be accepted. The limitations of frequentist analysis can also be problematic in fMRI analysis as well. In many cases, researchers are interested in examining whether neural activity in voxels is significantly different across different conditions (Han, 2020b). In other words, the presence of a significant non-zero effect in voxels becomes the main interest. In this situation, p-values can only provide information regarding whether a null hypothesis, “the effect in voxels of interest is zero,” can be rejected, not regarding whether an alternative hypothesis, “there is a non-zero effect in voxels,” is supported by data and thus can be accepted. Instead, the result from Bayesian analysis can directly suggest whether the alternative hypothesis is more favored by evidence compared with the null hypothesis that the most researchers are likely to be interested in Han, Park & Thoma (2018). Thus, the aforementioned epistemological benefit of Bayesian analysis that it can directly examine the alternative hypothesis and provide information about whether data supports the hypothesis is applicable to fMRI analysis as well.

Bayesian analysis has been implemented in several tools for fMRI analysis (Han, in press; Han & Park, 2018; Mejia et al., 2020). For instance, SPM provides options for performing Bayesian analysis in its analysis modules (Han & Park, 2018). In addition, several software tools have also been developed to implement more customized Bayesian analysis procedures. For example, BayesFactorFMRI provides options for performing meta-analysis of statistics images created by previous fMRI studies and Bayes factor-applied second-level analysis with multiple comparison correction with a graphical user interface and multiprocessing (Han, in press). BayesfMRI implements the Bayesian general linear model for fMRI data. In the present study, we will briefly review the conceptual basis of Bayesian analysis, particularly use of Bayes factors for inference, and consider how to improve its performance in the context of fMRI analysis (Mejia et al., 2020).

Bayesian fMRI analysis using Bayes factors

Bayesian analysis is mainly concerned with discovering the posterior distribution of a parameter of interest through the observation of collected data (Wagenmakers et al., 2018b; Han, Park & Thoma, 2018). On the other hand, frequentist analysis is usually more focused on examining the extremity of the observed data distribution given a hypothesis about the parameter of interest, which has been indicated by a p-value (Cohen, 1994). Unlike the frequentist approach, Bayesian analysis begins with a prior distribution of the parameter of interest; this prior distribution can be informed by a priori information (informative prior) or not (non-informative prior) (Stefan et al., 2019). This distribution is updated through iterative observations of data. Let us assume that we are interested in examining a hypothesis, H, with data, D (Marsman & Wagenmakers, 2017). Then, the updating occurs based on Bayes’ Theorem: (1) PH|D=PD|HPHPD

where P(H|D) is a posterior distribution of H that is updated from a prior distribution of H, P(H), with P(D|H), the probability of observing D given H, and P(D), a marginal probability, which is a constant for normalization (Han & Park, 2018). While frequentist analysis is more concerned about P(D|H), the likelihood that the observed data is the case given the hypothesis, Bayesian analysis can better inform us whether our hypothesis is likely to be valid given the observed data, P(H|D), that we are more interested in the most cases.

From a Bayesian perspective, testing an alternative hypothesis of interest (H1), such as presence of a non-zero effect in a specific voxel (e.g., activity in condition A > condition B), can be performed by comparing the posterior probability of the alternative hypothesis and that of the respective null hypothesis (H0) (Han, 2020b). With Bayes’ Theorem in (1), the comparison of these two posterior distributions can be done as follows: (2) PH1|DPH0|D=PD|H1PH1PDPD|H0PH0PD=PH1PH0×PD|H1PD|H0

The second term in the right-hand side of (2), PD|H1PD|H0, is a Bayes Factor (BF) that indicates the ratio of the amount of evidence supporting H1 versus H0. In this case, the calculated BF can be written as BF10. In the same way, if the null hypothesis, H0, becomes the main hypothesis of interest, then BF01=PD|H0PD|H1 that indicates the ratio of evidence supporting H0 versus H1 could be examined.

A BF quantifies to what extent the present evidence supports an alternative hypothesis (H1) versus a null hypothesis (H0) (in the case of BF10) or vice versa (in the case of BF01) (Kass & Raftery, 1995). BF10 greater than 1 indicates that the evidence favorably supports an alternative hypothesis while if it is smaller than 1, that indicates that evidence favors a null hypothesis (Wagenmakers et al., 2018a). For instance, BF10 = 10 indicates that evidence supports H1 ten times more strongly than H0. According to the general guidelines, BF10 ≥ 3 indicates presence of evidence positively supporting H1 versus H0, BF10 ≥ 6 indicates presence of strong evidence, and BF10 ≥ 10 indicates presence of very strong evidence. If BF10 is smaller than 3, then the evidence is deemed to be anecdotal, not decisive (Kass & Raftery, 1995).

One of the most fundamental epistemological benefits of using BFs in lieu of p-values in inference is that use of BFs allows us to more directly examine an alternative hypothesis (Wagenmakers et al., 2018b; Han, Park & Thoma, 2018). Let us assume that we intend to test whether a specific voxel reported a significant non-zero effect. In this case, H0 is about absence of a non-zero effect while H1 is about presence of a non-zero effect. In most cases, we are more interested in testing the presence of the non-zero effect (H1) instead of the absence of the effect (H0). Unfortunately, p-values are mainly concerned about P(D|H), the probability of observed data given the hypothesis. Thus, significant p-values, such as p < .05, can only inform us about the extremity of the data distribution given the hypothesis and whether H0 is likely to be rejected. In other words, p-values could not be used to make the decisions about whether H1 should be accepted in lieu of H0 that most of us are usually interested in our experiments. Instead, BFs are based on P(H|D), the probability of the hypothesis given observed data, so BFs allow us to directly examine to what extent our hypothesis of interest, H1 is supported by data compared with H0. The previous studies that attempted to employ BFs in fMRI analysis also pointed out that BFs can be used to more directly test whether voxels show significant activity H1 instead of merely testing H0 (Han & Park, 2018; Han & Park, 2019; Han, 2020b).

Importance of prior distribution selection

Although BFs possess the aforementioned practical benefits in inference, their appropiate use requires one fundamental issue to be addressed. Previous studies in Bayesian statistics have shown that a change in the prior distribution, P(H), could significantly alter the resultant BF (Liu & Aitkin, 2008; Sinharay & Stern, 2002). In order to address this issue and provide general guidelines for prior selection, several researchers have proposed use of default prior distributions, such as the Cauchy distribution with a scale (σ) 12=.707 (Gronau et al., 2017; Rouder & Morey, 2012). Previous Bayesian fMRI analysis studies that employed BFs also used such a default prior distribution (Han & Park, 2019; Han, 2020b). However, given that BFs are sensitive to the change in the prior distribution, the use of the default prior could not be ideal in all instances (Sinharay & Stern, 2002).

As a possible solution, several researchers have attempted to create a prior distribution from the results of relevant previous studies (Van de Schoot et al., 2018; Zondervan-Zwijnenburg et al., 2017). Given that the prior distribution is about the likelihood distribution of the parameter of interest prior to observing data (Marsman & Wagenmakers, 2017), which is based on our prior knowledge, according to its definition, it would be plausible to utilize previous findings to formulate the distribution to address issues that might emerge from arbitrary prior selection (Zondervan-Zwijnenburg et al., 2017). In the fields of clinical psychology and data science, the performance of the prior distribution informed by relevant previous studies and meta-analyses has been tested in several previous studies (Van de Schoot et al., 2018; Avci, 2017; Zondervan-Zwijnenburg et al., 2017). They have shown that the use of prior distributions that were properly designed with a priori information resulted in better performance in terms of the smaller variability in predicted trend lines and lower deviance information criterion, which indicates the production of a better regression model, compared with when the default prior distributions were used. The methodological implications of these previous findings suggest that use of an informative prior distribution that is appropriately informed by findings from relevant previous studies and meta-analyses can improve the performance of analysis and help us feasibly address the aforementioned issue related to the selection of prior distributions (Zondervan-Zwijnenburg et al., 2017; Van de Schoot et al., 2018).

Present study

To examine whether Bayesian analysis with a Cauchy prior distribution informed by a priori information can also improve performance in fMRI research as shown in the previous non-fMRI studies, outcomes from different analysis methods were compared in the present study. The compared analysis methods included Bayesian analysis with the Cauchy prior distribution informed by a priori information, Bayesian analysis with the default Cauchy prior, and frequentist analysis with voxelwise family-wise error (FWE) correction implemented in SPM 12.

In the present study, a computational method to adjust the scale of a Cauchy prior distribution based on a priori information that can be extracted from relevant previous fMRI studies, was proposed. Although several previous studies have employed BFs in Bayesian fMRI analysis (Han & Park, 2018; Han, 2020b; Han, in press), they relied upon the default Cauchy prior distribution, so it would be necessary to consider how to properly determine the prior distribution based on information. Thus, how a Cauchy prior distribution should be adjusted based on results from previous studies and meta-analyses was considered. The proposed method was used to adjust the Cauchy prior distribution, which was employed in voxelwise Bayesian second-level fMRI analysis with BFs, by adjusting its scale, σ, with a priori information. Then, whether use of the adjusted Cauchy prior distribution with the adjusted σ resulted in the improvement of performance in terms of sensitivity and selectivity was examined. For the planned performance evaluation, three different sets of fMRI images, one consisting of images created by simulations and two consisting of real brain activity contrast images from previous fMRI studies, were analyzed.

Methods

Use of a priori information from previous studies to adjust prior distributions

In this section, I will consider how to determine an appropriate prior distribution for Bayesian fMRI analysis based on a priori information. In particular, I will focus on Bayesian analysis using BFs for inference based on observed data. Given that a change in the selected prior distribution can significantly alter the resultant BFs in inference (Liu & Aitkin, 2008; Sinharay & Stern, 2002), it is necessary to carefully consider how to choose an appropriate prior distribution. Hence, based on the idea that use of a priori information from relevant previous studies in designing a prior distribution produced better analysis outcomes (Van de Schoot et al., 2018; Zondervan-Zwijnenburg et al., 2017), I will explore how to apply such an approach in the context of fMRI analysis in the present study.

The previous studies that used BF-based Bayesian fMRI analysis have utilized a Cauchy prior distribution (Han, 2020b; Han, in press). The Cauchy distribution has been widely used as a prior distribution in BF-applied Bayesian analysis in the field (Gronau et al., 2017; Rouder & Morey, 2012). This distribution is determined by two parameters, a center location (x0) and scale (σ). x0 determines where the peak of the distribution will be located. σ determines the width of the distribution; an increase in σ results in a wider distribution (Sahu et al., 2019). For illustrative purposes, Fig. 1 demonstrates several Cauchy distributions with different x0s and σs.

Figure 1 Example of Cauchy prior distributions.

For general purposes, such as t-tests and ANOVA, Gronau et al. (2017) proposed the default Cauchy prior distribution. The default Cauchy prior distribution uses σ=12=.707 (Gronau et al., 2017). This prior distribution has been widely used in tools that implement BF-applied Bayesian analysis, such as JASP and BayesFactorFMRI (Han, in press; Marsman & Wagenmakers, 2017). Although this has become a well-recognized default prior distribution in the field, given that use of an informative prior distribution informed by relevant a priori information resulted in better analysis outcomes (Van de Schoot et al., 2018; Zondervan-Zwijnenburg et al., 2017), it would be worth considering how to adjust a Cauchy prior distribution informed by such information. Thus, as a possible solution, a method to adjust σ based on the results of relevant previous fMRI studies, particularly their meta-analysis, was developed and tested in the present study. Once we meta-analyze statistics image files that were created by the previous studies with the image-based meta-analysis method, information required for the prior adjustment can be acquired. For instance, if we are interested in comparing neural activity between two conditions of interest, the meta-analysis of relevant previous studies can inform us regarding the difference in activity across conditions found in the previous studies.

Again, assume that we are interested in testing whether there is a significant difference in neural activity between two conditions with a BF in our planned analysis. Then, we will examine the resultant BF10 that indicates to what extent the collected data supports our alternative hypothesis, H1, that there is a significant difference in neural activity between conditions of interest versus the null hypothesis, H0. In order to calculate BF10 in this case, we need to start with defining the prior distribution, P(H0). It would be possible to approximate the shape of the distribution if we have a priori information. If we intend to acquire the information from the aforementioned meta-analysis, then we can utilize these three types of information: first, the difference in neural activity across two conditions or contrasts; second, noise strength; and third, the proportion of significant voxels.

The aforementioned contrast directly influences the overall shape of the prior distribution, P(H0), particularly in terms of its width. Consider two hypothetical cases. When the contrast value reported from the meta-analysis is large, we are likely to expect that we observe a large contrast and more significant voxels from the planned analysis. On the other hand, if the meta-analysis reported a small contrast value, then it could be predicted that a small contrast and less significant voxels are likely to be found from the planned analysis. Given that these expectations constitute our prior brief and then the basis of the prior distribution, the prior distribution is likely to be strongly centered around x0 = 0 with a steep peak (e.g., Cauchy(x0 = 0, σ = .25) in Fig. 1) when the estimated contrast value is small. If the expected contrast value is greater, then the peak at x0 = 0 should be less steep and the overall distribution should be more dispersed (e.g., Cauchy(x0 = 0, σ = 2) in Fig. 1). Similarly, noise strength can also influence the overall shape of the prior distribution. As the estimated noise strength increases, the expected effect size decreases when a difference (or contrast) is set at a constant value (e.g., Cohen’s D=mean differencestandard deviation). Thus, the prior distribution has a steeper peak at x0 = 0 as the noise strength estimated from the meta-analysis increases given that the estimated effect size is inversely proportional to the noise strength and we expect to see fewer significant voxels at the end. Finally, the proportion of significant voxels that survived thresholding in the prior studies also influences the prior determination process. If there were more voxels that showed significant non-zero effects (e.g., activity in condition A > condition B) and survived thresholding, more significant effects are expected to be found in the planned analysis. Accordingly, the prior distribution to be used is likely to show a less steep peak at x0 = 0 in this case compared with when the proportion of significant voxels is smaller.

Consider how to adjust σ in a Cauchy prior distribution based on the aforementioned a priori information, the contrast (C), noise strength (N), and proportion of significant voxels (R). C is the contrast in terms of the difference in mean activity strength between significant versus non-significant voxels. N is the strength of noise, such as the standard deviation of the reported activity strength in voxels. R represents the ratio of significant voxels out of all analyzed voxels; for instance, R = 1.60% means that 1.60% of all analyzed voxels were significant in the example. Based on the information, it is possible to estimate the expected effect in each of the potentially significant voxels. This value becomes greater as C increases but becomes smaller as N increases. Also, if more voxels were found to be significant in the prior analysis, in other words if R is greater, then we can expect the expected effect value to become greater. Hence, the expected effect value can be estimated as: (3) X=CNR

Cauchy prior distribution adjustment

In general, comparing activity in voxels between two conditions is the main objective in fMRI analysis. In many cases, researchers are interested in whether the activity in a task condition is significantly greater than that in a control condition. To adjust a prior distribution in these cases, the estimated X from (3) can be used to determine x in a probability distribution as a candidate for a threshold. Once we consider the cumulative probability of a prior distribution, we may assume that the cumulative probability between −∞ and X, Pr[ − ∞ ≤ x < X], becomes a specific amount, P. For instance, once we assume that P = 95%, then at X, the cumulative probability becomes Pr−∞≤x<X= ∫−∞Xfxdx=95%. Based on these assumptions, we can set a specific P for the prior distribution to be used. The percentile, P, in the scale adjustment process can be estimated as: (4) P= ∫−∞Xfxdx

where f(x) is a Cauchy distribution, Cauchy(x0 = 0, σ), with σ to be determined. This P determines the shape of the Cauchy prior distribution. If X is constant, then the greater P results in a smaller σ and a Cauchy distribution with a steeper peak at x0. Once we expect that more incidences are situated <X (greater P), then there should be more incidences at x0 (greater probability density) where the peak is. For instance, let us compare two cases, P = 95% and P = 80% (see Fig. 2 for an illustration). As demonstrated in the figure, setting P = 95% resulted in the Cauchy distribution with the steeper peak at x0 = compared with setting P = 85% when X = .032.

Figure 2 Cauchy prior distributions with different Ps.

Based on the determined P and X, with (4), we can numerically search for σ in order to adjust the Cauchy prior distribution. Here is one illustrative example. Let us consider a case when 100,000 voxels are analyzed. To acquire information to calculate X, it is required to know C, N, and R. Let us assume that a meta-analysis of relevant previous studies indicated that a total of 1,600 voxels were active, while the difference between the mean activity strength between significant versus non-significant voxels was 1.0, and the standard deviation of the activity strengths, the noise strength, was .50. In this case, C = 1.0, N = .50, and R=1,600100,000=1.60%. Based on these assumptions, X=CNR=1.0.501.60%=.032. If we intend to find a Cauchy distribution scale σ that suffices P = 95%, then we need to find a σ that suffices: 95%= ∫−∞.032Cauchyx0=0,σdx

Once we numerically search for a σ that suffices in the equation above, then we can find that σ ≈ .0051 in this case. By using the same approach, we can calculate σs for different Ps, such as 80%, 85%, and 90%. When we calculate the aforementioned σs, they become .023, .016, and .010, respectively. When all other parameters, C, N, and R, are the same, the Cauchy prior distribution with the adjusted σ becomes more centered around zero with a steeper peak as P increases. Figure 2 demonstrates the different Cauchy prior distributions with different Ps when the parameters used in the prior example were applied. In general, as P increases, the resultant Cauchy distribution becomes more concentrated around zero. In general, use of a narrower Cauchy prior distribution centered around zero with the steeper peak resulting from a greater P is likely to produce a more stringent result in terms of fewer voxels that survived thresholding.

Adjusting a cauchy prior distribution with information from meta-analysis of relevant previous fMRI studies

With (3) and (4), it is possible to adjust σ to generate a customized Cauchy prior distribution if prior information for C, N, and R is available. For instance, if we meta-analyze relevant previous studies with resultant images from the studies, we can estimate the proportion of voxels that survived thresholding (R), the mean activity strength difference in significant voxels (positives) versus non-significant voxels (negatives) (C), and the overall noise strength in terms of the standard deviation or activity strengths (N).

From the meta-analysis, C can be estimated by comparing the mean of the activity strength in significant voxels that survived thresholding versus that in non-significant voxels. When the activity strength difference in each significant voxel i is Dv1i, that of each non-significant voxel j is Dv0j, the number of significant voxels is nv1, and that of non-significant voxels is nn0, C can be estimated as follows: (5) C=∑Dv1inv1 ∑Dv0jnv0

N can be estimated from the standard deviation of the activity strengths in the whole image, i.e., SD(Dv), when Dv is the activity strength difference in each of all voxels in the result of the meta-analysis, including both significant and non-significant voxels. Finally, R, indicating the ratio of the significant voxels in the whole resultant image, can be calculated as follows: (6) R=nv1nv1+nv0

Once all these parameters are determined, with (3), X can be calculated. A specific P value can be chosen, but in this study, I will test the performances of P = 80%, 85%, 90%,  and 95%. With X and P, σ of the adjusted Cauchy prior distribution to be used for analysis can be found from (4) numerically. Then, the adjusted Cauchy(x0 = 0, σ) can be used as the prior distribution in further fMRI analysis.

Performance evaluation with simulated images

To examine the performance of the adjusted prior distribution, analysis of simulated images was conducted, as in Han (2020b). A series of images with different proportions (.01% to 25.60%) of true positives was generated. For instance, in the case of the simulated image with 25.60% true positives, the image contained sphere-shaped true positives and the total number of voxels that contained the true positives was 25.60% of the whole image. The intensity of the voxels with true positives was 1, while all other voxels without true positives were coded as 0. As a result, the contrast of the simulated active vs. inactive voxels was set to 1 (C = 1) in the present study. A total of twelve simulated images with twelve different true positive proportions, R = .01%, .02%, .05%, .10%, .20%, .40%, .80%, 1.60%, 3.20%, 6.40%, 12.80%, and 25.60%, were generated. Examples of the generated images (R = .10%, .20%, .40%, .80%) are presented in the left hand side of Fig. 3 (Left).

Figure 3 Examples of the created simulated images (C = 1.00, R = .10%, .20%, .40%, .80%).

Note. Left: original simulated images (without the random noise). Right: images with the added random noise, N(.00, .50).

As a way to examine whether the tested analysis methods were reliable in second-level fMRI analysis, false positive (FPR) and hit rates (HR) were used as performance indicators (Han & Park, 2018; Han, 2020b). The FPR indicates how many voxels are found to be significant after thresholding despite the voxels are not true positives. The HR refers to the extent to which an analysis method can properly detect truly significant voxels after thresholding. A better performing analysis method reports a lower FPR, which is related to Type I error, and a higher HR, which is related to Type II error. To examine FDPR and HR, noise-added images were analyzed and the analysis results were compared with the original simulated images containing true positives. In the present study, a series of noise-added images were generated by adding the random noise following Gaussian distribution, N(0, .50), to the original images (See the right hand side of Fig. 3 for examples). For each of the aforementioned twelve proportion conditions, noise-added images were created with four different sample sizes, n = 8, 12, 16,  and 20. For each proportion and sample size condition, ten different sets of images were created to repeat testing. For instance, in the case of the proportion condition, R = .01%, ten sets of noise-added images generated for each of four different sample size conditions.

For the evaluation for performance, FDPR and HR were used in the present study. With the original and noise-added images, FPR and HR can be calculated as follows (Han & Park, 2018; Han, 2020b): (7) FPR=nFalsePositivenNoisePositive

(8) HR=nTruePositivenOriginalPositive

where nFalsePositive is the number of voxels that are reported to be significant from analysis but not actually positive in the original image; nNoisePositive is the number of voxels that are reported to be significant from analysis; nTruePositive is the number of voxels that are reported to be significant from analysis and also actually positive in the original images; and nOriginalPositive is the number of voxels that are actually positive in the original image. For the interpretation of the evaluation results, the analysis method that reported lower FPR and higher HR was deemed as the better method. FPR was utilized as a proxy for selectivity and HR was utilized as a proxy for sensitivity.

The noised-added images were analyzed with the Bayesian second-level analysis with adjusted Cauchy prior distributions generated by the method explained previously. One-sample Bayesian t-tests were performed with Python and R codes modified from BayesFactorFMRI (Han, in press) (see https://github.com/hyemin-han/Prior-Adjustment-BayesFactorFMRI for the source code and data files). The resultant images were thresholded at BF ≥ 3 for evaluation (Kass & Raftery, 1995; Han, Park & Thoma, 2018). In addition, the multiple comparison-corrected default Cauchy prior distribution, which had σ = .707 before correction, was also employed. The multiple comparison-corrected default Cauchy prior distribution was generated with the method proposed by Han (2020b) and De Jong (2019), which adjusts σ based on the number of voxels to be tested. The use of the corrected default Cauchy prior was intended to examine whether the proposed method for prior adjustment can produce relatively improved performance. Similarly, the resultant images were also thresholded at BF ≥ 3. Simultaneously, the same images were also analyzed with the frequentist analysis method with the voxelwise FWE correction implemented in SPM 12 for the performance comparison (Han & Glenn, 2018). The images were entered into the second-level analysis model for one-sample t-tests in SPM 12. Then, the results were thresholded at p < .05 (voxelwise FWE corrected).

Performance evaluation with concrete examples: working memory fMRI

In addition to the examination of the performance with the simulated data, the performance of the Bayesian fMRI analysis method was also tested with real case examples of two working memory fMRI datasets. The performance was evaluated in terms of the extent to which the result of the Bayesian analysis and thresholding overlapped with the results from the large-scale meta-analyses of relevant previous fMRI studies (Han & Glenn, 2018; Han, Glenn & Dawson, 2019). The working memory fMRI datasets were analyzed with the proposed Bayesian analysis method with the prior adjustment. σ for the prior adjustment was determined with meta-analysis of relevant previous studies.

For the second-level fMRI analysis for performance evaluation, contrast images in the working memory fMRI datasets were used. The contrast images that contained results from the first-level (individual-level) analysis of fMRI images from the working memory experiments were obtained from publicly available fMRI datasets. The images in the first dataset were generated from the analysis of fMRI images acquired from fifteen participants. In the original study, DeYoung et al. (2009) examined the neural activity while participants were solving three-back working memory task problems. The contrast images at the individual level were obtained by calculating the difference between the neural activity during the control condition and that during the task condition (Task − Control) for each participant. Further details regarding the data acquisition and analysis processes are described in DeYoung et al. (2009). The fifteen contrast images are available via a NeuroVault repository by Kragel et al. (2018), https://neurovault.org/collections/3324/ (Study7 files). Each downloaded contrast image contained the contrast of the neural activity in the working memory task condition versus the control condition.

The second dataset consisted of contrast images collected from thirteen participants. This dataset was collected in a previous study done by Henson et al. (2002) that compared neural activity in the memory recall task versus control conditions. Similar to DeYoung et al. (2009), one contrast image was generated for each participant. Further details about the experiment and analysis procedures are available in Henson et al. (2002). The contrast images were downloaded from the SPM tutorial dataset repository (https://www.fil.ion.ucl.ac.uk/spm/download/data/face_rfx/face_rfx.zip; files under “cons_can” folder). Because the files were in the ANALYZE format, fslchfiletype was performed to convert the files into the NIfTI format.

Bayesian second-level analysis was performed with the aforementioned contrast images containing results from first-level (individual-level) analyses in the two previous working memory fMRI studies (DeYoung et al., 2009; Henson et al., 2002). Similar to the case of the analysis of the simulated images, a one-sample t-test was performed (see https://github.com/hyemin-han/Prior-Adjustment-BayesFactorFMRI for the source code and data files). While conducting Bayesian second-level fMRI analysis, the Cauchy prior was adjusted with information from image-based meta-analysis of statistical images containing either t- or z-statistics reported from previous fMRI studies of working memory. The meta-analyzed statistics images were downloaded from NeuroVault, an open online repository to share fMRI statistics images (Gorgolewski et al., 2015). The descriptions of the individual studies included in the meta-analysis are presented in Table 1. The meta-analysis was performed with the results from six analyses reported in five previous studies (Kiyonaga, Dowd & Egner, 2017; Quinque et al., 2014; Demetriou et al., 2018; Egli et al., 2018; Stout et al., 2017). Bayesian meta-analysis of the statistics images from the previous studies was performed with BayesFactorFMRI (Han, in press).

Table 1 Previous fMRI studies included in the meta-analysis.

Authors	Year	Sample size	Compared task conditions	NeuroVault Collection/Image ID	
Quinque et al.	2014	18	(Remembered + forgotten) vs. null events	145/781	
Kiyonaga et al.	2017	28	Hard vs. easy search	1354/19037	
Demetriou et al.	2018	14	2-back vs. 0-back working memory tasks	1922/29310	
Demetriou et al.	2018	14	2-back vs. 0-back working memory tasks	1922/29328	
Egli et al.	2018	1,369	2-back vs. 0-back working memory tasks	2621/50291	
Stout et al.	2017	81	2-face vs. 1-face working memory tasks	2884/53141	

From the meta-analysis, the parameters required to adjust σ were calculated. The proportion of voxels found to be significant in terms of BF ≥ 3 was R = 12.92%. The mean activity strength in the significant voxels was .61 and that in the non-significant voxels was .06, so C = .55. Finally, the noise strength was calculated from the standard deviation of the activity strength in all analyzed voxels, so N = .33. With these parameters, the calculated σs for the adjusted Cauchy prior distributions when P = 80%, 85%, 90%,  and 95% were .16, .11, .07, and .03, respectively. Bayesian second-level analysis was performed with the module for one-sample t-tests implemented in BayesFactorFMRI (Han, in press). For evaluation with FDPR and HR, resultant images were thresholded at BF ≥ 3 (Han, Park & Thoma, 2018; Kass & Raftery, 1995). The thresholded results were stored in binary images; each voxel in the images indicated whether the voxel was significant (1) or not (0) after thresholding.

Furthermore, as was done with the simulated images, the downloaded images were also analyzed with the corrected default Cauchy prior distribution and frequentist FWE correction implemented in SPM 12. Bayesian second-level analysis with the corrected default Cauchy prior was performed by BayesFactorFMRI (Han, in press). The resultant images were thresholded at BF ≥ 3. In addition, similar to the aforementioned Bayesian second-level analysis, SPM 12 was used to perform a one-sample t-test at the group level. Once the analysis process was completed, the resultant statistics image was thresholded at p < .05 with voxelwise FWE correction implemented in SPM 12. For the evaluation of the performance, the thresholded results were stored in binary images as done previously.

The performance in this case was evaluated in terms of the quantified overlap between the resultant thresholded images and the results of large-scale meta-analysis of the relevant previous fMRI studies. In the present study, an overlap index, which ranges from 0 (no overlap) to 1 (complete overlap), was calculated by using the formula proposed by Han & Glenn (2018). An overlap index between the original and target images, I, can be calculated as follows: (9) I=2|Vovl||Vovl||Vorg||Vtar||Vovl||Vorg|+|Vovl||Vtar|

where Vovl is the number of voxels that were significant in both original and target images, Vorg is that of the significant voxels in the original image, and Vtar is that of the significant voxels in the target image. For instance, if 800 voxels were significant in both images, while a total of 1,600 voxels were significant in the Bayesian analysis result image (original image) and 2,000 voxels were significant in the meta-analysis result image (target image), I can be calculated as: I=2|Vovl||Vovl||Vorg||Vtar||Vovl||Vorg|+|Vovl||Vtar|=2800160080020008001600+8002000=.44

In the present study, the resultant images from the performed analyses were used as the original images to be examined and the images that were generated from the large-scale meta-analyses were used as the target images. For the evaluation, images generated from meta-analyses of previous fMRI studies of working memory have been used as the target images in the present study. First, statistics images that reported results from relevant prior studies were downloaded from NeuroVault and were meta-analyzed with the Bayesian fMRI meta-analysis tool in BayesFactorFMRI (Han & Park, 2019; Han, in press). Second, a resultant image from the coordinate-based activation likelihood estimation meta-analysis of neuroimaging data was employed. Activation foci information was acquired from BrainMap using Sleuth that enables users to explore the large-scale database containing coordinate information in previously published fMRI papers (Laird, Lancaster & Fox, 2005). The keyword “working memory” was used to search for the previous studies relevant to the present study. Once the activation foci information was acquired, the information was entered into GingerALE for meta-analysis (Eickhoff et al., 2009; Eickhoff et al., 2012; Turkeltaub et al., 2012). The resultant image from GingerALE reported the voxels showed significant common activity in the working memory task conditions when the cluster-forming threshold p < .001 and cluster-level FWE threshold p < .01 were applied. Third, the result of the meta-synthesis of previous fMRI studies about working memory was also used. An image was obtained from NeuroSynth (Yarkoni et al., 2011), an online tool for meta-synthesis of neuroimaging studies, with the keyword ”working memory.” The downloaded image reported voxels that showed significant activity when the likelihood of activity in ”working memory” given that in all possible task conditions, P (working memory | all conditions), was examined. Fourth, a result from the automated meta-analysis of neuroimaging studies implemented in NeuroQuery was utilized (Dockès et al., 2020b). NeuroQuery “is focused on producing a brain map that predicts where in the brain a study on the topic of interest is likely report observations (Dockès et al., 2020a).” A NeuroQuery map was created with the keyword “working memory” and downloaded.

Vovls were calculated with the thresholded images from our analyses (original images) and the aforementioned four meta-analysis results (target images). For the original images, thresholded images generated by BayesFactorFMRI (for Bayesian analysis) and SPM 12 (for frequentist analysis) were employed. Four different meta-analysis result images were used for the target images. While interpreting the resultant Vovl, the higher value was assumed as the indicator for the better performance.

Results

Performance evaluation with simulated images

We examined the FPRs and HRs across different conditions with the simulated images that contained sphere-shaped true positives. Figure 4 demonstrates how the FDPRs Left; A to D) and HRs (Right; E to H) changed with the change in the proportion of the true positives (R) across different sample sizes (n) and analysis methods. In the case of subfigures reporting FPRs, black horizontal lines that represent FPR = .05 were added for reference.

Figure 4 False positive and hit rates evaluated with the simulated images.

. (A-D) False positive rates with different sample sizes (N = 8, 12, 16, 20). (E) to (H) Hit rates with different sample sizes (N = 8, 12, 16, 20).

In the cases of the FPRs (the left hand side of Figs. 4A–4D), the classical frequentist FWE reported the lowest rates, which were always lower than the criterion level, .05, in all instances. In general, the FPRs resulting from Bayesian analysis became lower as the sample size, N, increased. The FPRs resulting from Bayesian analysis with the default Cauchy prior distribution monotonically decreased as the proportion of true positives, R, increased. A similar pattern was found from Bayesian analysis with the adjusted prior distributions. However, FPRs in these cases showed the second peaks around R = 6.40% to 25.60%. FPRs decreased as the higher, more stringent P, was applied

HRs (the right hand side of Figs. 4E–4H) increased as the sample size, N, increased. In all instances, HRs resulting from the classical frequentist FWE and Bayesian analysis with the default Cauchy prior distribution did not change significantly across different Rs. When Bayesian analysis with the adjusted priors was performed, HRs increased as R increased. The lower, more lenient P resulted in a relatively higher HR when the other parameters were the same.

Performance evaluation with concrete examples: working memory fMRI data

The analysis results from different methods are presented in Fig. 5 (analysis of DeYoung et al., 2009) and Fig. 6 (analysis of Henson et al., 2002). Similar to the cases of the analyses of simulated images, first, the classical frequentist voxelwise FWE correction resulted in the least significant voxels, which represent the stringency of the thresholding methods. Second, Bayesian analysis with the default Cauchy prior distribution after correction produced the less significant voxels than Bayesian analysis with the adjusted Cauchy prior distributions. Third, fewer voxels were significant as the higher P was employed in Bayesian analysis with the adjusted Cauchy prior distributions.

Figure 5 Results of analysis of the working memory fMRI images with different methods (DeYoung et al., 2009).

Note. Red: significant voxels in each thresholding condition (A) Classical frequentist voxelwise FWE. (B) Bayesian analysis with the corrected default Cauchy prior. (C) Bayesian analysis with the adjusted Cauchy prior with P = 80% (D) P = 85%. (E) P = 90%. (F) P = 95%.

Figure 6 Results of analysis of the working memory fMRI images with different methods (Henson et al. 2002).

Note. Red: significant voxels in each thresholding condition (A) Classical frequentist voxelwise FWE. (B) Bayesian analysis with the corrected default Cauchy prior. (C) Bayesian analysis with the adjusted Cauchy prior with P = 80% (D) P = 85%. (E) P = 90%. (F) P = 95%.

For the evaluation, we examined the indices of overlap between the resultant images and meta-analysis images as references. The overlap indices resulting from the comparisons are reported in Table 2 (analysis of DeYoung et al., 2009) and Table 3 (analysis of Henson et al., 2002). In the comparisons with all different types of meta-analysis images, Bayesian analysis with the adjusted Cauchy prior distributions showed better outcomes compared with Bayesian analysis with the default Cauchy prior distribution after correction as well as classical frequentist voxelwise FWE correction.

Discussion

In the present study, we developed and tested a method to create a Cauchy prior distribution for Bayesian second-level fMRI analysis by adjusting its distribution scale parameter, σ, with information acquired from relevant previous studies. The performance of the method was tested by comparing it with that of classical frequentist voxelwise FWE correction (Han & Glenn, 2018) and that of Bayesian analysis with the default Cauchy prior distribution with multiple comparison correction (Han, 2020b; De Jong, 2019). The performance evaluation was conducted with both the simulated images and real image datasets, the working memory fMRI images (DeYoung et al., 2009; Henson et al., 2002). We demonstrated that it would be possible to adjust σ based on a priori information, which can be retrieved from analysis, particularly meta-analysis, of relevant previous studies. The required information includes the proportion of significant voxels in the whole image (R), the contrast in terms of the difference between the mean activity in significant voxels versus non-significant voxels (C), and the noise strength, such as the standard deviation of the activity strengths of voxels, (N).

Table 2 Overlap indices resulting from the comparisons between meta-analysis images and thresholded images (DeYoung, 2009).

	Bayesian meta-analysis	brainmap + GingerALE	NeuroSynth	NeuroQuery	
Bayesian: 80%	.23	.25	.20	.16	
Bayesian: 85%	.23	.26	.21	.16	
Bayesian: 90%	.22	.27	.24	.18	
Bayesian: 95%	.22	.27	.24	.18	
Bayesian: Default	.09	.14	.17	.13	
Classical	.00	.00	.00	.00	
Notes.

Rows indicate different analysis methods applied. Columns indicate different types of meta-analyses used for comparisons.

Table 3 Overlap indices resulting from the comparisons between meta-analysis images and thresholded images (Henson, 2002).

	Bayesian meta-analysis	brainmap + GingerALE	NeuroSynth	NeuroQuery	
Bayesian: 80%	.12	.09	.05	.06	
Bayesian: 85%	.12	.09	.05	.06	
Bayesian: 90%	.12	.08	.05	.05	
Bayesian: 95%	.12	.08	.04	.05	
Bayesian: Default	.08	.06	.04	.05	
Classical	.01	.01	.00	.02	
Notes.

Rows indicate different analysis methods applied. Columns indicate different types of meta-analyses used for comparisons.

When the performance was evaluated with the simulated images, in all instances, Bayesian analysis in general showed the better sensitivity in terms of the HR compared with classical frequentist analysis. This result was consistent with a previous study that examined the performance of Bayesian analysis with the corrected default Cauchy prior distribution (Han, 2020b). Although the classical frequentist voxelwise FWE correction reported the lowest FPR below .05, its HR was always significantly lower than the HR of Bayesian analysis. With a relatively larger sample size (n ≥ 12), Bayesian analysis with the adjusted Cauchy prior distribution reported a desirable level of FPR, FPR < .05. The reported FPR was lowest when P = 95% was employed. On the other hand, the reported HR increased as the more lenient P, such as 80%, was employed. In order to examine the relative superiority and inferiority of each method, I will refer to these criteria:

1. FPR should be lower than .05.

2. When 1 is satisfied, HR should be .75 or higher

3. If there is no case that satisfies both 1 and 2, compare HRs between the cases that satisfy at least 1.

Based on these criteria, the performance of each analysis method in each condition, in terms of the sample size (n) and proportion of the a priori true positives (R), was evaluated and presented in Fig. 7 (A to D according to n). For this purpose, the simulated images were analyzed and tested. As shown, in all cases except one case when n = 12 (B) and R = .01%, Bayesian analysis was superior to classical frequentist voxelwise FWE correction because it showed the relatively higher HR even when n and R were small. When Bayesian analysis was conducted with versus without the adjustment of σ in the Cauchy prior distribution, the application of the prior adjustment resulted in better performance, particularly when the proportion of the a priori true positives (R) was small. When R was small, the application of the default Cauchy prior distribution resulted in an increased FPR greater than .05, so it would not be suitable to control for false positives when the true positives are assumed to be rare in the analyzed images. In addition, the difference in the employed Ps produced the difference in the performance outcomes. In general, as mentioned previously, the lower P resulted in the higher HR; however, when n and R were small, the employment of low P, such as P = 80% or 85% tended to result in the unacceptable FPR, FPR ≥ .05. In those cases with the small n and R, the relatively better results were reported when P = 95% was applied. In several instances with n ≥ 12 (B) and small R (e.g., .05% ≥ R ≥ .20% when n = 12 and R ≤ .02% when n = 16), better outcomes were achieved with P = 90% than P = 95% due to the higher HR; however, this trend diminished as R increased.

Figure 7 Comparison of different methods based on the three evaluation criteria.

(A) N = 8. (B) N = 12. (C) N = 16. (D)N = 20. Black: FPR ≥ .05. Blue: FPR < .05 and HR ≥ .75. Cells with numbers: cases when FPR < .05 but HR < .75. The numbers indicate the relative superiority of each method. 1 indicates the best method within the given n and R.

In addition, use of the adjusted Cauchy prior distribution can improve performance in terms of the overlap with results from large-scale meta-analyses when the real image datasets are analyzed. The method was tested with the working memory fMRI datasets containing fifteen contrast image files from DeYoung et al. (2009) and Henson et al. (2002). The performance of each analysis method was tested by examining the extent to which the results of the analyses overlapped the results of large-scale meta-analyses conducted with Bayesian image-based meta-analysis (Han & Park, 2019), GingerALE (Eickhoff et al., 2009; Eickhoff et al., 2012; Turkeltaub et al., 2012), NeuroSynth (Yarkoni et al., 2011), and NeuroQuery (Dockès et al., 2020b). In all cases, Bayesian second-level fMRI analysis performed with the adjusted Cauchy prior distribution reported higher overlap indices than either Bayesian analysis performed with the corrected default Cauchy prior distribution or classical frequentist voxelwise FWE correction. This result suggests that the proposed method for the prior adjustment may improve performance even when real image files, not hypothetical simulated image files, are analyzed.

In the present study, a method to determine σ for the adjustment of the Cauchy prior distribution in Bayesian second-level fMRI analysis was proposed and tested. If prior information about estimated true positives is available, σ can be determined and the prior distribution can be adjusted accordingly as suggested. In general, similar to the case of Bayesian analysis with the corrected default Cauchy prior distribution (Han, 2020b), the proposed method resulted in significantly better sensitivity in terms of HR compared with frequentist voxelwise analysis with FWE correction, which has been reported to be very selective but very conservative (Lindquist & Mejia, 2015). In addition, when the adjusted Cauchy prior was used, the reported performance was better than when the default Cauchy prior corrected for multiple comparisons was used, particularly when the sample size and the proportion of true positives were small. This result is consistent with previous non-fMRI studies that compared the performance of the default prior versus that of the prior informed by relevant previous literature (Van de Schoot et al., 2018; Avci, 2017; Zondervan-Zwijnenburg et al., 2017). Given these results, the proposed method for the adjustment of σ and the Cauchy prior distribution will be able to contribute to the improvement of the performance of Bayesian analysis in fMRI research. Because one benefit of employing Bayesian analysis is that it is possible to use information retrieved from relevant previous studies to properly design the prior distribution to be used in the current research project, the present study would provide useful insights about how to feasibly apply the aforementioned idea in Bayesian fMRI analysis. In addition, as the source code files based on BayesFactorFMRI were shared via GitHub (https://github.com/hyemin-han/Prior-Adjustment-BayesFactorFMRI), an open repository, fMRI researchers who intend to use the proposed method will be able to easily test it.

The application of Bayesian analysis with an adjusted prior distribution can contribute to solving a contemporary statistical issue about reliability in fMRI analysis. There have been increasing concerns regarding whether fMRI research can show acceptable reliability and validity (Zuo, Xu & Milham, 2019; Elliott et al., 2020). For instance, a recent study reported that the analyses of large-scale fMRI datasets showed poor test-retest reliability (Elliott et al., 2020). Given that one of major sources of the poor reliability and validity is the random noise (Zuo, Xu & Milham, 2019), the method proposed in the present study can potentially provide researchers with one possible way to alleviate the issue. As shown in the analyses of simulated images, Bayesian analysis with an adjusted prior demonstrated relatively consistent outcomes in terms of FPRs and HRs even with the presence of the random noise. Future studies should test whether the proposed method can actually improve the reliability of fMRI analysis within the context of task-based fMRI in addition to second-level fMRI analysis, which has been examined in the present study.

However, several limitations in the present study warrant further investigation. First, the same adjusted Cauchy prior distribution is to be applied in all voxels as a global prior, so it could not take into account any voxel-specific or local-level factors in the prior adjustment process. Because the provision of a simple and feasible method for prior adjustment with minimal a priori information was the main purpose in the present study, a more sophisticated method that allows the consideration of voxel-specific or local-level aspects should be examined in future research. Second, because we set one of the parameters required for σ determination, P, prior to the analysis, it may cause the issue of arbitrary prior determination. Although P = 95% showed the best performance in the most cases, further research should be done to explore the best way to systematically determine P. Third, for optimal determination of parameters based on meta-analysis, image-based meta-analysis instead of coordinate-based meta-analysis, which is more frequently utilized in the field, should be performed. This could be a potentially significant issue due to the lack of open statistical images files for image-based meta-analysis available to the public. NeuroVault is one of the repositories to share such image files (Gorgolewski et al., 2015), but the limited availability of statistical images resulting from diverse experiments should be addressed in the long term.

Additional Information and Declarations

Competing Interests

Author Contributions

Data Availability

The author declares that they have no competing interests.

Hyemin Han conceived and designed the experiments, performed the experiments, analyzed the data, prepared figures and/or tables, authored or reviewed drafts of the paper, and approved the final draft.

The following information was supplied regarding data availability:

Source code and data files are available at GitHub, link: https://github.com/hyemin-han/Prior-Adjustment-BayesFactorFMRI/.

The meta-analyzed statistics images are also via a NeuroVault repository (https://neurovault.org/collections/3324/).

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
