# Peer review of "A method to adjust a prior distribution in Bayesian second-level fMRI analysis"

_PeerJ, doi:10.7717/peerj.10861_

## Round 0.1 · original submission · Major Revisions

Your manuscript has now been seen by two reviewers. You will see from their comments below that while they find your work of interest, some major points are raised. We are interested in the possibility of publishing your study, but would like to consider your response to these concerns in the form of a revised manuscript before we make a final decision on publication. We therefore invite you to revise and resubmit your manuscript, taking into account the points raised. Please highlight all changes in the manuscript text file.

·

Basic reporting

1. The paper requires complete restructuring/reorganization. Sentences used are quite long and confusing. For example, in line number 7 to 10, the sentence is vague with sub-strings “In terms of…” used twice in one sentence. Such style makes it hard to follow the paper. There are multiple issues like this.
2. Starting from line 13 to 15 “Such as the result from relevant previous studies…” seems to be out of context in a sentence.
3. The paper starts with the keyword fMRI in the title itself and throughout the paper. Please add a full form of it on its first occurrence in the abstract/introduction. I believe this would increase the readability of paper for a general audience. Likewise, there are several other instances where the full form of the acronym is not present.
4. In the same line’s introduction should start with a paragraph discussing the wider subject area.
5. In the line 51, the first sentence of the paragraph should be supported with at least one or two citations as it claims that Bayesian analysis is implemented in a wide range of tools.
6. In the line number 251, the last sentence at page number 7 can mislead. “Figure 3 (Left)” can be interpreted as something which is on the left side.
7. Figure 4 covers a major part of the performance evaluation study done by the author. Results also show improvement in terms of iterate and false positive. However, there is no in detail explanation provided or all eight sub-figures (part of figure 4). This way it is all left to the reader to interpret. I strongly advise explaining quantitative and qualitative trend to show in at least two to three paragraphs.
8. Along the same lines, explanations are needed for figure 5 and figure 6. It is even more important when these figures are referred.
9. Section 4 (discussion section) starts with the sentence “In the present paper…”. That sentence can be framed in a better way.
10. In the line number 410 citation is incorrectly presented.
11. There are multiple instances of grammatical errors. For instance, Line number 456: “It may suggest that…” should be changed to “It is suggested that…”.

Experimental design

The paper proposes a method based on Cauchy prior distribution to analyse Bayesian second level fMRI images. The research area has the potential to directly impact society and therefore relevant. This is an incremental work` and performance results are promising. It would be even better if more arguments are presented to support the result obtained.

Validity of the findings

The obtained results are validated and conclusion is well stated.

Additional comments

Please make corrections as suggested.

·

Basic reporting

Figures 5-6: The colors (red, blue, sky blue) did not show up correctly from the reviewer's side. I can see yellow and white as well as tiny dots in black or dark blue. Please make sure of this issues addressed in the revision.

Experimental design

No comment.

Validity of the findings

No comment.

Additional comments

The present work extends previous developed Bayes Factor-applied second-level fMRI analysis by adjusting the Cauchy prior distribution based on a priori information derived from relevant previous studies. The authors reported the improvement of the performance of proposed Bayesian second-level fMRI analysis. This is an interesting direction of increasing the validity of fMRI-based approaches. While all the tests in the present work focus on group-level characteristics, such an improvement could also be effective on individual differences. Recently, task-based fMRI has been argued with limited test-retest reliability of measurements on the inter-individual differences (PMID: 32489141). A demonstration of the improvements on the task-based fMRI reliability using this method would be highly recommended. More details of the reliability crisis in neuroscience can be found in a recent comment (PMID: 31253883).

---

## Round 0.2 · accepted · Accept

Thank you for the response letter and revised version of the manuscript. We are delighted to accept your manuscript for publication.

·

Basic reporting

The author has incorporated all the suggestions in the revised paper.

Experimental design

Now improved the presentation. All suggestions have been incorporated in the revised manuscript.

Validity of the findings

Good. Revised version looks fine to me.

Additional comments

All suggestions have been incorporated in the revised manuscript.
A good work.

·

Basic reporting

no comment

Experimental design

no comment

Validity of the findings

no comment

Additional comments

The revision has been quite addressing the issues raised.